# Individual Deviation-Based Functional Hypergraph for Identifying Subtypes of Autism Spectrum Disorder

**DOI:** 10.3390/brainsci14080738

**Published:** 2024-07-24

**Authors:** Jialong Li, Weihao Zheng, Xiang Fu, Yu Zhang, Songyu Yang, Ying Wang, Zhe Zhang, Bin Hu, Guojun Xu

**Affiliations:** 1Gansu Provincial Key Laboratory of Wearable Computing, School of Information Science and Engineering, Lanzhou University, Lanzhou 730000, China; 220220942951@lzu.edu.cn (J.L.); fux2019@lzu.edu.cn (X.F.); yzhang20@lzu.edu.cn (Y.Z.); 220220942481@lzu.edu.cn (S.Y.); wangying22@lzu.edu.cn (Y.W.); 2Institute of Brain Science, Hangzhou Normal University, Hangzhou 311121, China; zhangz@hznu.edu.cn; 3School of Physics, Hangzhou Normal University, Hangzhou 311121, China; 4School of Medical Technology, Beijing Institute of Technology, Beijing 100081, China; 5CAS Center for Excellence in Brain Science and Intelligence Technology, Shanghai Institutes for Biological Sciences, Chinese Academy of Sciences, Shanghai 200031, China; 6Joint Research Center for Cognitive Neurosensor Technology of Lanzhou University & Institute of Semiconductors, Chinese Academy of Sciences, Lanzhou 730000, China; 7Key Laboratory for Biomedical Engineering of Ministry of Education, Department of Biomedical Engineering, College of Biomedical Engineering & Instrument Science, Zhejiang University, Hangzhou 310027, China

**Keywords:** machine learning, autism spectrum disorder, heterogeneity, diagnostic information, individual deviation, hypergraph community detection, reproducible subtypes

## Abstract

Heterogeneity has been one of the main barriers to understanding and treatment of autism spectrum disorder (ASD). Previous studies have identified several subtypes of ASD through unsupervised clustering analysis. However, most of them primarily depicted the pairwise similarity between individuals through second-order relationships, relying solely on patient data for their calculation. This leads to an underestimation of the complexity inherent in inter-individual relationships and the diagnostic information provided by typical development (TD). To address this, we utilized an elastic net model to construct an individual deviation-based hypergraph (ID-Hypergraph) based on functional MRI data. We then conducted a novel community detection clustering algorithm to the ID-Hypergraph, with the aim of identifying subtypes of ASD. By applying this framework to the Autism Brain Imaging Data Exchange repository data (discovery: 147/125, ASD/TD; replication: 134/132, ASD/TD), we identified four reproducible ASD subtypes with roughly similar patterns of ALFF between the discovery and replication datasets. Moreover, these subtypes significantly varied in communication domains. In addition, we achieved over 80% accuracy for the classification between these subtypes. Taken together, our study demonstrated the effectiveness of identifying subtypes of ASD through the ID-hypergraph, highlighting its potential in elucidating the heterogeneity of ASD and diagnosing ASD subtypes.

## 1. Introduction

Autism spectrum disorder (ASD) is an increasingly common neurodevelopmental condition characterized by impairments in social communication, emotional functioning, and restricted and repetitive behaviors (RRB) [1,2]. *The Diagnostic and Statistical Manual of Mental Disorders*, Fourth Edition, revised (DSM-IV), refers to ASD as pervasive developmental disorder, including childhood autism, Asperger syndrome, Rett’s syndrome, childhood disintegrative disorders, and pervasive developmental disorders to be categorized [3], whereas the Fifth Edition of DSM (DSM-V) eliminated the concepts of these syndromes and defined all of them as autism spectrum disorders [4]. However, the clinical phenotypes of individuals with ASD are highly heterogeneous, and the underlying neural mechanism is unclear [5,6,7]. Previous studies have found impairments in the autistic brain [8,9,10], and remarkable performance in classifying ASD and typical development (TD) was achieved based on machine learning approaches [11,12,13,14]. Though important, these findings showed limited reproducibility across combined samples from different data sources [15,16]. A predominant reason for this phenomenon lies in the heterogeneity [17,18], which may lead to less reproducibility of statistical analyses and poor diagnostic accuracy [19]. Thus, identifying biologically driven subtypes within ASD may contribute to elucidating the biological mechanisms of heterogeneity in autistic brains and potentially facilitate the clinical diagnosis of ASD subtypes.

Recent studies have made significant progress in dissecting the heterogeneity of ASD through unsupervised clustering analysis, such as k-means [20], hierarchical clustering [21,22], and non-negative matrix factorization [23], which aim to directly find the optimal subtypes of patients based on their neuroimaging features. In addition, community detection clustering algorithms were also applied to subtyping ASD [24,25]. Indeed, regardless of neuroimaging modality and methodology, these studies converged to reveal at least two-four subtypes of ASD. Nevertheless, these algorithms primarily identified subtypes through the distance or second-order relationship between paired samples, which may struggle to capture the high-order relationships among multiple samples. How to precisely represent the biological similarity among samples remains a challenge. The hypergraph model serves as a high-order structure adept at precisely encoding high-order relationships among multiple samples [26,27]. Recently, the hypergraph constructed based on sparse representation (e.g., lasso) has been introduced into brain functional network studies [28,29]. Actually, the relationships among individuals with ASD may also exhibit a high-order structure. Therefore, we hypothesize that the hypergraph might be a more appropriate structure for modeling the high-order relationships, potentially allowing for a finer elucidation of the biological mechanisms of heterogeneity in autistic brains.

Although significant progress has been made by previous studies [20,21,22,23,24,25,30,31,32,33], most of them focused solely on the ASD population for dissecting the biological subtypes of ASD, ignoring the inter-individual deviation across the ASD and TD populations. These patient-focused subtyping approaches tend to ignore the fact that the brain-level variability of the ASD population often overlaps with the TD population [34], which significantly hampers the reveal of a complete picture of ASD biology [35]. It becomes imperative to incorporate diagnostic information to subtype and profile the deviations between the ASD and TD populations. Hence, we proposed a novel subtyping approach by focusing on inter-individual deviation with ASD and characterized the high-order relationships among inter-individual deviation with ASD by establishing an ID-Hypergraph through the elastic net model. On the other hand, there is currently a notable abundance of clustering algorithms designed for hypergraphs, such as hypergraph spectral clustering [36], hypergraph clustering based on modularity feature projection [37], and the hypergraph maximum-likelihood Louvain [38]. It is crucial for delineating the heterogeneity of the ASD population.

In this paper, we aim to address two issues: (1) how to represent high-order relationships of inter-individual deviation between ASD and TD groups and (2) how to decompose the community architecture of ID-Hypergraph. Here, we constructed an ID-Hypergraph using the elastic net model based on fMRI data. This model incorporated L1-norm and L2-norm designed to address the grouping effect while selecting all relevant individuals with ASD. Then, we utilized a novel hypergraph community detection clustering algorithm for decomposing the community architecture of the ID-Hypergraph. We evaluated the proposed method by analyzing data from 147 ASDs and 125 TDs sourced from the discovery dataset and validated its reproducibility based on the replication dataset. The support vector machine (SVM) classifier was utilized to examine the separability and rationality of the identified subtypes. In addition, we analyzed the differences between the ASD and TD groups to support the existence of deviation between both. Furthermore, we characterized the subtypes-specific changes in brain manifestations and symptoms by comparing ALFF and clinical assessments of each subtype against the TD group, as well as between the subtypes.

The contributions of this paper can be summarized below.

We proposed an individual deviation-based hypergraph (ID-Hypergraph) model, which characterizes high-order relationships among individuals, to parse the neuroactivational heterogeneity of ASD;We identified four ASD subtypes with heterogeneous changes in both brain activity and behavior domains;The identified ASD subtypes were highly separable and were reproducible across different datasets.

## 2. Materials

### 2.1. Participants

We utilized the fMRI and phenotypic data downloaded from the ABIDE database (http://fcon_1000.projects.nitrc.org/indi/abide/, accessed on 7 February 2021) [39,40]. We analyzed two sets of multisite neuroimaging data: I) ABIDE-I (Discovery, 147/125, ASD/TD) and II) ABIDE-II (Replication, 134/132, ASD/TD) datasets. Data acquisition at each site was approved by the local Institutional Review Board. Participants with ASD were diagnosed by experienced clinicians using the Autism Diagnostic Observation Schedule (ADOS) scores. Participants were excluded according to the following criteria: (1) female participants; (2) participants categorized as left-handed, mixed-handed, or those lacking handedness information; (3) participants with head motion over 1.0 mm translation or 1.0 degree; and (4) study sites with less than 10 participants with ASD. The demographic information of the remaining participants is summarized in Table 1.

### 2.2. Image Preprocessing

Preprocessing of the fMRI data was conducted via the Data Processing Assistant for Resting-State fMRI (DPARSF) toolbox in MATLAB software (the version number: R2020a). The preprocessing steps included removing the first 10 time points, slice time correction, head motion correction, registration, spatial normalization to the standard MNI space with a voxel size of 3 × 3 × 3 mm, spatial smoothing, and band-pass filtering (0.01–0.1 Hz). The influences of nuisance variables (gray matter, white matter, cerebrospinal fluid, and global signals) were regressed. The mean ALFF of each brain region was calculated according to the AAL template.

## 3. Methods

Figure 1 illustrates our proposed novel subtyping framework based on the ID-Hypergraph model, which incorporates diagnostic information provided by the TD population. This framework identifies subtypes of ASD based on the inter-individual deviation between ASD and TD populations. Briefly, the mean ALFF of the brain regions was extracted from the preprocessed images, and the cosine distance was utilized to calculate the IDALFF (inter-individual deviation) according to the ALFF of ASD and TD group, which were subsequently used to estimate the high-order relationships among inter-individual deviation through the elastic net model. This process constructed an ID-Hypergraph specific to the IDALFF. Then, the hypergraph community detection clustering strategy was applied to identify the subtypes of ASD.

### 3.1. Preliminaries on Hypergraph

Denote a hypergraph G=(V,E,w) and its vertex set $V=[v_{1},v_{2},\cdots ,v_{N}]$, hyperedge set $E=[e_{1},e_{2},\cdots ,e_{M}]$ with ${⋃}_{j=1}^{M}e_{j}=V$, and hyperedge weight vector $w={\left[w\left(e_{1}\right),w\left(e_{2}\right),\cdots ,w\left(e_{M}\right)\right]}^{T}\in R^{M}$, where each $e_{j}$ is assigned with a weight $w(e_{j})$. We can represent G using a |V|×|E| incidence matrix H with the following elements:(1)$$H\left(v,e\right)=\left\{\begin{array}{lll} 0, & if & v\notin e\\ 1, & if & v\in e \end{array}\right.$$
when $v_{i}$ ∈ $e_{j}$, i.e., $H_{ij}=1$, a hyperedge $e_{j}$ connects with a vertex $v_{i}$.

Based on the incidence matrix H, the degree of vertices and hyperedges could be defined as
(2)$$d\left(v_{i}\right)=\sum_{e_{j}\in E}w\left(e_{j}\right)H_{ij}\;\;\;\mathrm{for}\;\;\;1\leq i\leq N$$
(3)$$\delta \left(e_{j}\right)=\sum_{v_{i}\in V}H_{ij}\;\;\;\mathrm{for}\;\;\;1\leq j\leq M$$

Let $D_{v}$ and $D_{e}$ denote the diagonal matrices of the degree of vertices and hyperedges, respectively, i.e., $D_{v}=diag\left(d\left(v_{1}\right),d\left(v_{2}\right),\cdots ,d\left(v_{N}\right)\right)\in R^{N\times N}$ and $D_{e}=diag\left(\delta \left(e_{1}\right),\delta \left(e_{2}\right),\cdots ,\delta \left(e_{M}\right)\right)\in R^{M\times M}$; W denotes the matrix of hyperedge weights, i.e., $W=diag(w)=diag\left(w\left(e_{1}\right),w\left(e_{2}\right),\cdots ,w\left(e_{M}\right)\right)\in R^{M\times M}$. Here, we simply set W as an identity matrix.

### 3.2. Construction of Inter-Individual Deviation-Based Hypergraph

#### 3.2.1. Inter-Individual Deviation of ALFF (IDALFF)

The mean ALFF feature vector was extracted for each subject as a measure of brain region. The IDALFF of subject *i* from the ASD group is calculated by
(4)$$X_{ij}=E\left\{1-\frac{F_{i}F_{j}^{\prime }}{\sqrt{\left(F_{i}F_{i}^{\prime }\right)\left(F_{j}F_{j}^{\prime }\right)}}\right\},j=1,2,\ldots m$$

$F_{i}$ denotes the overall ALFF of subject *i* in the ASD group. Similarly, $F_{j}$ denotes the overall ALFF of subject *j* in the TD group. To characterize the inter-individual deviation, the cosine distance of the overall ALFF between each subject *i* in the ASD group and all subjects in the TD group were calculated to characterize the IDALFF of subject *i*. The IDALFF of all subjects in the ASD group were aggregated to form the IDALFF matrix ($X\in R^{n\times m}$).

#### 3.2.2. Inter-Individual Deviation Based Hypergraph (ID-Hypergraph)

We employed an elastic net model to estimate the high-order relationships based on inter-individual deviation. Specifically, ${X=[x_{1},\ldots ,x_{i},\ldots ,x_{n}]}^{T}\in R^{n\times m}$ is the IDALFF matrix collected from each subject, where *n* and *m* are the numbers of subjects and inter-individual deviation, respectively. For the $X\in R^{n\times m}$, we calculated the incidence matrix $H\in R^{n\times n}$ via the elastic net model by minimizing the following objective function [41]:(5)$$\underset{w}{\min}\sum_{i=1}^{n}{\left\|x_{i}-X_{i}w\right\|}_{2}+\lambda_{1}{\left\|w\right\|}_{1}+\lambda_{2}\;{\left\|w\right\|}_{2}^{2}\;\;\;\;\;\;\;for\;\;\;1\leq i\leq n$$
where ${X_{i}=[x_{1},\ldots ,x_{i-1},0,\ldots x_{i+1},\ldots ,x_{n}]}^{T}\in R^{n\times m}$ is the inter-individual deviation of all the subjects except the *i*-th subject, and $w\in R^{n\times 1}$ is the coefficient vector of hyperedge. The centroid subject *i* and those subjects with the non-zero coefficient in the weight vector w were included for a hyperedge. In addition, $\lambda_{1}>0$ is a regularization parameter controlling the sparsity of the model; a larger $\lambda_{1}$ leads to a sparser w and vice versa. The $\lambda_{2}>0$ is the regularization parameter controlling the group selection of the model. In addition, we utilized a grid search to determine the optimal regularization parameters, thereby constructing the hypergraph with a relative optimal sparsity. The hyperedges of all centroid subjects were aggregated to form an incidence matrix ($H\in R^{n\times n}$), serving as the ID-Hypergraph.

### 3.3. Hypergraph Community Detection

We utilized the Hypergraph Maximum-Likelihood Louvain (HMLL) [38] to optimize the symmetric hypergraph modularity, which performed approximate maximum likelihood inference via coordinate ascent. Let $Z_{i}\in [\bar{C}]=\{1,2,\ldots ,\bar{C}\}$ denote the cluster membership of node i and collect these assignments in a vector Z. The symmetric modularity objective of hypergraph is formulated as
(6)$$Q\left(Z\right)=-\sum_{k=1}^{\bar{k}}\beta_{k}\left[{cut}_{k}(Z)+\gamma_{k}\sum_{C=1}^{\bar{C}}vol(C{)}^{k}\right]$$

The $\bar{k}$ is the maximum degree value of the hyperedge. The parameter $\beta_{k}$ is used to control the optimal size of hyperedges for clustering, and $\gamma_{k}$ controls the size of clusters. The partition term, which computes the number of hyperedges of size k that contain nodes in two or more distinct clusters, is formulated as
(7)$${cut}_{k}\left(Z\right)\equiv m_{k}-\sum_{R\in R^{k}}a_{R}\delta \left(Z_{R}\right)$$
where $m_{k}=\sum_{R\in R^{k}}\;a_{R}$. The term $\sum_{C=1}^{\bar{C}}\;vol({C)}^{k}$ calculates the degree values of the nodes in all clusters.

The hypergraph community detection clustering algorithm was utilized to identify distinct subtypes of individuals with ASD. The algorithm that iteratively sorts the nodes into clusters until the hypergraph modularity $Q\left(Z\right)$ in Equation (6) reaches a maximum to find the optimal cluster membership Z, which is utilized to represent the final subtyping result.

### 3.4. SVM Classifier

We implemented the SVM classifier with the radial basis function kernel using the Scikit-learn toolbox [42]. A ten-fold cross-validation strategy was applied to search the optimal parameter C in the range of {2^−5^, 2^−4^, …, 2^1^, 2^2^} on the dataset. In addition, we used accuracy (ACC), sensitivity (SEN), specificity (SPE), and area under the receiver operating characteristic (ROC) curve (AUC) as indices for performance assessment, defined as
(8)$$\begin{array}{ll} ACC & =\frac{TP+TN}{TP+FN+TN+FP}\\ SEN & =\frac{TP}{TP+FN}\\ SPE & =\frac{TN}{TN+FP} \end{array}$$
where TP and TN represent the number of true positive and true negative, respectively. FP and FN denote the number of false positive and false negative, respectively.

### 3.5. Statistical Analysis

We analyzed the differences between the ASD and TD groups, as well as between each identified ASD subtype and the TD group. A two-sample *t*-test was utilized to assess group differences on whole-brain ALFF and clinical symptoms (ADOS scores), with age and site as covariates. The results were corrected using false discovery rate (FDR) correction [43].

### 3.6. Reproducibility Analysis

All analyses were repeated using the replication dataset. Notably, to test the reproducibility of each identified subtype, the exact clustering parameters and clusters used in the discovery dataset were reused. The characterization of the subtypes identified from the replication dataset was then re-evaluated to quantitatively assess their reproducibility and generalizability.

## 4. Results

### 4.1. Altered ALFF between ASD and TD Group

The differences between the ASD and TD groups were analyzed using a two-sample *t*-test. In the case-control comparison, distinct alteration patterns of ALFF were observed. As shown in Figure 2, ASD showed significant increases of ALFF in the orbitofrontal, supplementary motor, rectus, and middle temporal gyrus and decreases in calcarine, lingual gyrus, and thalamus when compared to the TD group (FDR *q* < 0.05, Figure 2). Therefore, there are significant differences between the ASD and TD groups, as well as some overlap (no significant difference). The result in Figure 2 can provide evidence for the existence of inter-individual deviation between the ASD and TD groups. We hypothesize that utilizing inter-individual deviation between the ASD and TD groups may reveal the full extent of biological heterogeneity in ASD.

### 4.2. Subtyping ASD Based on ID-Hypergraph

We hypothesized the existence of ASD subtypes with high heterogeneity of functional brain characteristics, and clustering analysis was performed on 147 individuals with ASD and 125 individuals with TD, as illustrated in Figure 1. By using the subtyping framework based on the ID-Hypergraph model, we identified four subtypes of ASD. For the optimal value of the cluster, it is determined by the hypergraph modularity $Q\left(Z\right)$ in Equation (6). In addition, the same analysis was used for the replication dataset. The subtyping framework found four ASD subtypes roughly similar to the discovery findings. Then, we delved into a comprehensive profiling of their alterations in ALFF to test the reproducibility of the identified subtypes.

### 4.3. Classification Between ASD Subtypes

We further examined the separability of the identified subtypes by utilizing a classification task with a ten-fold cross-validation strategy. Regional ALFF were employed as features and input into the SVM classifier. The optimal hyperparameter C was determined to be 1.2 by searching through the grid search. To classify more than two subtypes, we established one SVM classifier for each pair of subtypes. The final multi-classification results were derived by averaging the outcomes of all the classifiers. The classification between subtypes achieved excellent diagnostic performance, with a mean classification accuracy of 80.74%, specificity of 82.29%, sensitivity of 81.53%, and AUC of 0.8045. These results further confirmed that the ASD subtypes identified by the proposed method are biologically separable. This could help improve the diagnosis of ASD subtypes. Detailed classification results between each of the two subtypes are presented in Table 2.

### 4.4. Characterization of the ASD Subtypes

Compared to the TDs, subtype 1 showed significant differences in the insula and some subcortical nuclei (e.g., amygdala, putamen, and pallidum) (FDR *q* < 0.05, Figure 3). In subtype 2, significant ALFF differences were observed in the orbitofrontal and superior frontal areas (FDR *q* < 0.05, Figure 3). Subtype 3 showed an increase in the insula, cingulate, hippocampus, and some subcortical nuclei (e.g., caudate, putamen, pallidum, and thalamus) and a decrease in cuneus, temporal, parietal, and occipital regions (FDR *q* < 0.05, Figure 3). Finally, no significant difference was found between subtype 4 and the TDs (*p* > 0.05, Figure 3). We also compared the ALFF between the ASD subtypes, which revealed multiple brain regions showing significant differences, which can be seen in Appendix A.

As shown in Figure 3 and Table 3, four subtypes of ASD exhibit different patterns and symptom severity. Specifically, subtype 1 showed only increased ALFF in several brain regions and slight clinical symptoms. In contrast, subtype 2 showed only decreased ALFF in several brain regions and slight clinical symptoms. In addition, subtype 3 showed the most complex patterns of ALFF and severe clinical symptoms in terms of communication, social, and behavior scores. Previous studies have shown that the temporal region is critical for language function [44]. In addition, this region involves several functions (e.g., social and emotional processing) [45]. Therefore, we speculated that the more severe clinical symptoms of subtype 3 may be associated with the reductions of ALFF in the temporal region. Notably, the identified subtype 4 showed no significant differences in brain patterns of ALFF but slight clinical symptoms. Taken together, the four identified subtypes have shown different patterns and clinical symptoms, which may provide a profile for the delineation of subtypes of clinical practice.

### 4.5. Clinical Symptoms of the ASD Subtypes

The clinical symptoms were quantified through the ADOS scores. Subjects with missing ADOS scores were excluded from the analysis. As shown in Table 3 and Figure 4, distinct clinical phenotypes were observed between subtypes. Specifically, the ADOS communication score of subtype 2 is significantly lower than that of subtype 3 (*q* < 0.05, FDR corrected). No significant differences were found between the subtypes in other ADOS sub-domains. The demographic information for the identified subtypes is also presented in Table 3; differences in age and IQ were also observed between the subtypes.

### 4.6. Reproducibility of the ASD Subtypes

In the replication dataset, four subtypes of ASD were found, which exhibit roughly similar alteration patterns of ALFF in comparison with the discovery dataset. Specifically, subtype 1 showed significant differences in the inferior frontal, insula, cingulate, superior temporal, and some subcortical nuclei (e.g., caudate, putamen, and pallidum) (FDR *q* < 0.05, Figure 5). In subtype 2, significant differences were observed in the precentral and rolandic areas (FDR *q* < 0.05, Figure 5). Subtype 3 showed an increase in the cingulate, hippocampus, parahippocampus, temporal, and some subcortical nuclei (e.g., caudate, putamen, pallidum, and thalamus) and a decrease in orbitofrontal, cuneus, parietal and occipital regions (e.g., SOG, MOG, and IOG). (FDR *q* < 0.05, Figure 5). No significant difference was also found between subtype 4 and the TDs (*p* > 0.05, Figure 5).

In conclusion, the subtypes derived from the discovery and replication datasets revealed roughly similar patterns. Specifically, subtype 1 showed an increase in the insula and some subcortical nuclei in both the discovery and replication datasets. Subtype 2 exhibited a decrease in different brain regions in both the discovery and replication datasets. Subtype 3 showed an increase in the cingulate, hippocampus, and some subcortical nuclei and a decrease in cuneus, parietal, and occipital regions in both the discovery and replication datasets. Finally, no significant differences were found between subtype 4 and the TDs in both the discovery and replication datasets. Taken together, our results demonstrate that the identified subtypes have distinct patterns, which were reproducible in the independent data.

## 5. Discussion

Biological heterogeneity remains a main barrier to the development of reliable diagnostic criteria for autism [46]. Despite considerable efforts to develop various subtyping approaches for dissecting the heterogeneity of ASD [20,21,22,23,24,25,30,31,32,33], purely patient-focused approaches may lack a component capable of directly introducing diagnostic information from the TD population and characterizing high-order relationships among inter-individual deviation. We hypothesized that combining diagnostic information could provide a more comprehensive characterization of the heterogeneity present in autistic brains. To address this, we proposed a novel subtyping approach based on an ID-Hypergraph model, which integrated the high-order diagnostic information and detected subtypes through the hypergraph community detection clustering algorithm. This algorithm is highly scalable and can detect interpretable high-order structures [38].

Our approach utilized cosine distance to learn inter-individual deviation and adopted the elastic net to estimate high-order relationships among individual based on their deviation. The L2-norm in elastic net performs well with many predictors that are highly correlated with the dependent variable, making it more robust to extreme correlations among predictors than lasso [47,48]. Specifically, the elastic net can select all ASD patients who were highly associated with the target individual (the dependent variable) rather than the lasso that only selects one of them [26,49,50]. This advantage of the elastic net largely improves the construction of the ID-Hypergraph [51]. However, a major limitation of this approach is the difficulty in determining specific optimal sparsities for multiple regression tasks separately. To address this issue, we used a grid search to find the optimal sparsity parameter ($\lambda_{1}$) through ten-fold cross-validation for each regression task. Another limitation was the exclusion of female patients. Numerous studies have shown that ASD is a neurodevelopmental disorder more prevalent in males than in females [52,53], leading to a much higher number of males with ASD than females. Moreover, a study has shown that diagnostic criteria based on the characteristics of males with ASD are not fully applicable to the diagnosis of females with ASD, suggesting that the behavioral phenotype of males with ASD is significantly different from that of females [54]. Furthermore, the sample size of female patients in the ABIDE database is much smaller than that of males. Therefore, we excluded all female subjects from our study. In the future, more females with ASD can be collected subsequently to parse the ASD heterogeneity of female subjects.

Previous clustering algorithms for dissecting heterogeneity of ASD have mainly been based on distance calculation or second-order relationships represented by graphs, such as k-means [20], hierarchical clustering [21,22], non-negative matrix factorization [23], and community detection clustering algorithms on graphs [24,25]. To our best knowledge, this study represents the first attempt to parse the heterogeneity of ASD through the high-order biological relationships among inter-individual deviations. We achieved over 80% diagnostic accuracy for the classification between these subtypes through a simple SVM classifier. This result confirmed the effectiveness of our subtyping approach and highlighted its significance in the clinical diagnosis of ASD subtypes. The present study takes a further step to show that the hypergraph model is an effective structure to represent the high-order relationships among inter-individual deviations. This can be attributed to the enhanced capabilities of the hypergraph, which is adept at precisely capturing multivariate correlations (beyond the pairwise correlation) [55] and encoding more complex relationships, thereby providing a more comprehensive representation of the data [56].

By using the proposed subtyping framework, we identified four reproducible subtypes of ASD, which was consistent with recent studies identifying four ASD subtypes [32]. The group differences predominately exhibited a combination of increases and decreases in ALFF. Significant differences in the four subtypes were observed mainly in the orbitofrontal, insular, cingulate, hippocampus, temporal, parietal, and occipital regions and some subcortical nuclei. These findings were partially in line with previous studies, including impairments in the insula [57], hippocampus [58], as well as subcortical nuclei, such as the amygdala and thalamus [59,60], in individuals with ASD. These brain regions were reported to be involved in multiple cognitive functions. For example, the insula plays a pivotal role in socio-emotional and speech processing [61,62], and the thalamus is involved in modulating high-order functions such as emotion and consciousness [63,64]. Moreover, previous studies have shown that the temporal region was critical for language function [44]. In addition, this region involves several functions (e.g., social and emotional processing) [45]. Therefore, we speculated that the more severe impairments of subtype 3 may be associated with the reductions of ALFF in the temporal region. In addition, our identified subtypes demonstrated reproducibility of the identified subtypes in the independent dataset, which demonstrated excellent generalizability of our proposed subtyping framework.

This study has several limitations. First, all female patients were excluded, which limited the investigation of how gender impacted the heterogeneity of ASD. Second, we utilized a fixed sparsity parameter to construct a hypergraph among inter-individual deviation and set the weight to 1, ignoring the possible changes in the sparsity and strength of the hypergraph. In addition, the multimodality MRI data may better characterize the heterogeneity of ASD; we will use multimodality MRI data and develop novel algorithms specifically for multilayer hypergraph clustering.

## 6. Conclusions

In conclusion, our work proposed a novel ID-Hypergraph-based clustering framework for subtyping ASD. We generated the ID-Hypergraph by using cosine distance and elastic net model with the L1-norm and L2-norm to improve the construction of the ID-Hypergraph. The hypergraph community detection clustering algorithm was then utilized to dissect the biological heterogeneity of ASD. We identified four highly reproducible subtypes of ASD with roughly similar alteration patterns in brain function between the discovery and replication datasets. Furthermore, subtyping led to an excellent inter-subtype diagnostic performance, with an accuracy of over 80%. These results proved the effectiveness of employing inter-individual deviation information and a hypergraph model to dissect the biological heterogeneity in ASD.

## Figures and Tables

**Figure 1 brainsci-14-00738-f001:**
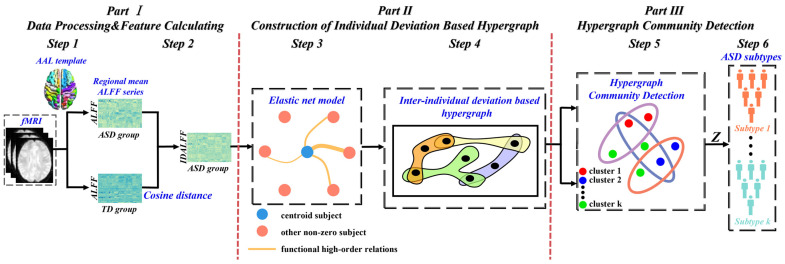
The flowchart of the subtyping framework based on the ID-Hypergraph model. (Part I) The mean ALFF was extracted from the preprocessed images, and the IDALFF (inter-individual deviation) was then calculated for each individual with ASD. (Part II) The estimation of high-order relationships among inter-individual deviation through the elastic net model. (Part III) The hypergraph community detection clustering algorithm for identifying subtypes of ASD.

**Figure 2 brainsci-14-00738-f002:**
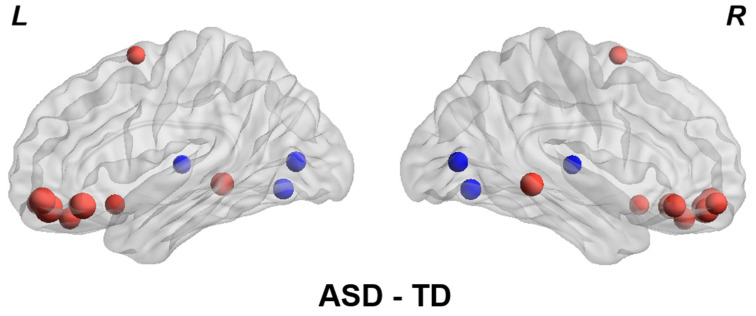
Comparison of ALFF between the ASD and TD groups. Group differences were calculated using two-sample *t*-test. Brain regions with FDR *q* < 0.05 were visualized. Red indicates that the ALFF value of this region is higher than that of TD, and blue indicates that the ALFF value of this region is lower than that of TD.

**Figure 3 brainsci-14-00738-f003:**
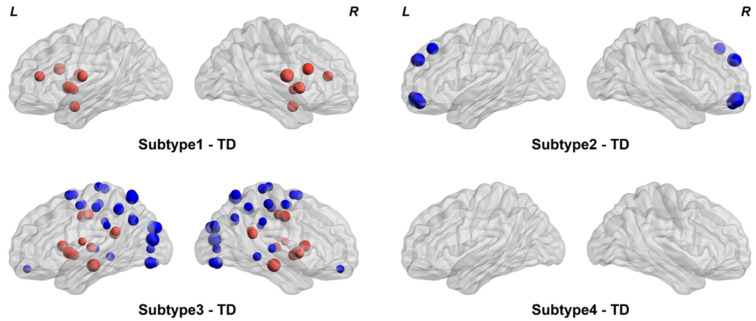
Comparisons of ALFF between each ASD subtype and the TDs. Group differences were calculated using two-sample *t*-test. Brain regions with FDR *q* < 0.05 were visualized. Red indicates that the ALFF value of this region is higher than that of TD, and blue indicates that the ALFF value of this region is lower than that of TD.

**Figure 4 brainsci-14-00738-f004:**
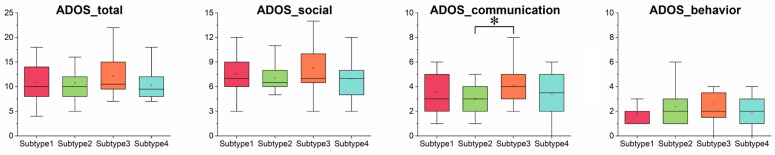
Comparisons of ADOS scores among subtypes of ASD. Group differences were assessed using two-sample *t*-test. * FDR *q* < 0.05. Abbreviations: ADOS, Autism Diagnostic Observation Schedule.

**Figure 5 brainsci-14-00738-f005:**
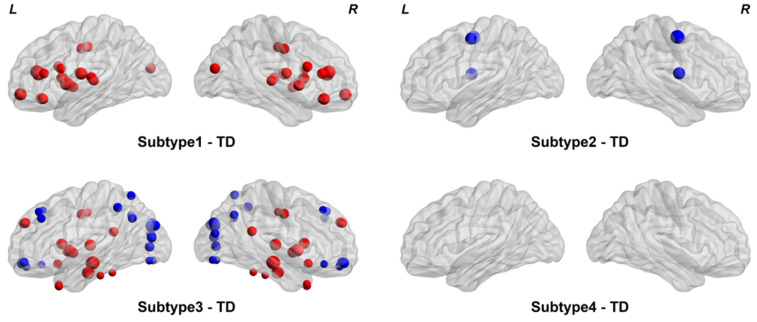
Comparisons of ALFF between each ASD subtype and the TDs in the replication dataset. Group differences were calculated using two-sample *t*-test. Brain regions with FDR *q* < 0.05 were visualized. Red indicates that the ALFF value of this region is higher than that of TD, and blue indicates that the ALFF value of this region is lower than that of TD.

**Table 1 brainsci-14-00738-t001:** Demographic information of remaining participants.

	Discovery (ABIDE-I)			Replication (ABIDE-II)		
Groups	ASD (n = 147)	TD (n = 125)	*p* Value	ASD (n = 134)	TD (n = 132)	*p* Value
Age	16.43 ± 8.15	17.73 ± 6.41	0.900 ^a^	15.72 ± 7.79	17.93 ± 6.28	0.326 ^a^
Gender	Male	Male	-	Male	Male	-
Handedness	Right	Right	-	Right	Right	-
IQ	110.75 ± 14.47	115.67 ± 12.74	0.009 ^a^	110.54 ± 16.96	114.27 ± 11.95	0.040 ^a^

Abbreviations: ASD, Autism Spectrum Disorder; TD, Typical Development; IQ, Intelligence Quotient; Values are mean ± standard deviation. ^a^ Two-sample *t*-test.

**Table 2 brainsci-14-00738-t002:** The detailed classification performance between each of the two subtypes.

Groups	ACC	SPE	SEN	AUC
Subtype1 vs. Subtype2	0.8643	0.8607	0.8333	0.8658
Subtype1 vs. Subtype3	0.8500	0.9267	0.7967	0.8467
Subtype1 vs. Subtype4	0.8125	0.7583	0.8233	0.8317
Subtype2 vs. Subtype3	0.7429	0.7517	0.7767	0.7075
Subtype2 vs. Subtype4	0.7428	0.7633	0.8017	0.7517
Subtype3 vs. Subtype4	0.8321	0.8767	0.8600	0.8233
Average result	0.8074	0.8229	0.8153	0.8045

ACC, SEN, SPE, and AUC are the abbreviations of ACCuracy, SENsitivity, SPEcificity, and the Area Under the receiver operating characteristic Curves, respectively.

**Table 3 brainsci-14-00738-t003:** Demographic and clinical information across subtypes.

Information	Subtype1	Subtype2	Subtype3	Subtype4
Age	23.33 ± 11.16	14.86 ± 3.97	18.41 ± 6.23	16.50 ± 7.27
IQ	107.70 ± 18.79	112.99 ± 11.71	113.13 ± 16.06	108.82 ± 12.38
ADOS				
Communication	3.57 ± 1.43	3.00 ± 1.18	4.06 ± 1.61	3.73 ± 2.00
Social	7.52 ± 2.60	7.07 ± 2.62	8.25 ± 3.07	7.45 ± 2.98
Behavior	1.83 ± 0.79	2.36 ± 1.43	2.56 ± 1.67	2.00 ± 1.26
Total	10.83 ± 3.87	10.78 ± 3.60	12.13 ± 4.35	11.18 ± 4.42

Abbreviations: IQ, Intelligence Quotient; ADOS, Autism Diagnostic Observation Schedule; Note: Values are mean ± std.

## Data Availability

We thank the numerous contributors to the ABIDE database for their efforts in collecting, organizing, and sharing their datasets. The data that support the findings of this study are openly available in the ABIDE at http://fcon_1000.projects.nitrc.org/indi/abide/, (accessed on 7 February 2021).

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
