# Peer review of "Individual Deviation-Based Functional Hypergraph for Identifying Subtypes of Autism Spectrum Disorder"

_brainsci, 2024, doi:10.3390/brainsci14080738_

Round 1

Reviewer 1 Report

Comments and Suggestions for Authors

In this study, the authors propose an ID-Hypergraph based on interindividual deviation to understand and treat ASD heterogeneity. The motivation of this model is to create hypergraphs based on individual deviation using functional MRI data and to detect autism subtypes through these hypergraphs. In the study, this framework was applied on the autism ABIDE dataset and four reproducible autism subtypes were examined. Additionally, the authors announced that classification success was achieved with over 80% accuracy among these subtypes.

1- The authors use a hypergraph model that takes into account higher order relationships for the first time in the literature to solve the heterogeneous structure of ASD.

2- Identifying autism subtypes with a classification accuracy of over 80% provides significant benefits in the diagnostic process.

3- The model contains generalizable features.

4- This study will enable the development of new approaches in the treatment and diagnosis of autism.

5- Can provide personalized treatments.

However, I kindly recommend that you take into consideration my gentle suggestions below.

1- The study was conducted only on male participants. This limits the study of the effect of gender on autism, meaning that study data must be analyzed with homogeneous data across all genders.

2- A fixed sparsification parameter was used in the hypergraph creation process and the weights were determined as 1. This means ignoring possible changes in the sparsity and strength of hypergraphs, which should be clarified separately in the article.

3- The study used only fMRI data. Multimodality MRI data should also be used.

4- Table 3 and Figure 4 should be compared together.

Reviewer 2 Report

Comments and Suggestions for Authors

In this paper, the authors present a novel approach using an individual deviation-based functional hypergraph (ID-Hypergraph) to identify subtypes of autism spectrum disorder (ASD).  This approach aims to better understand ASD heterogeneity and improve diagnostic accuracy. The researchers utilized fMRI data to construct the ID-Hypergraph and applied a community detection clustering algorithm to identify four reproducible ASD subtypes. Despite the proper use of the English language and a fine structure of the manuscript, some points need to be focused on improving the quality of the paper:

-       Improve the introductory section of the manuscript by highlighting and listing the “contributions'” points of this research;

-        

-       It is also necessary to explain better why a clustering approach is required for ASD problems in clinical practice.  Please highlight this critical aspect by comparing it with the actual clinical diagnosis;

-       The literature review is partial and incomplete, and some relevant ML approaches used as decision support systems should be cited and discussed: i.e., 10.3390/diagnostics11030574 (specific for ASD problem), then 10.1109/JPROC.2015.2501978, and 10.1007/s10462-017-9552-8.  I suggest putting them in the introductory section;

-       From a technical point of view, the authors should explore the impact of varying sparsity parameters on the construction of the ID-hypergraph to optimise model performance. Please answer and justify this in the manuscript.

-       The authors should investigate the integration of multimodal imaging data (e.g. structural MRI, DTI) to provide a more comprehensive analysis of ASD.

-       The Results section needs a proper comparison with other approaches with simple neural network architecture.  I recommend improving the paper by focusing on this crucial part of your case study.

Comments on the Quality of English Language

The overall quality of the manuscript's English can be enhanced, making it clearer and more professional.

Reviewer 3 Report

Comments and Suggestions for Authors

overall the manuscript is in good form, their are a few suggestions and comments that improve the quality of the manuscript.

1.     The introduction could be more concise by focusing on the most relevant background information and the research gap this study aims to fill.

2.     results section should provide a more in-depth interpretation of the findings, discussing the implications of the identified ASD subtypes and their potential relevance to clinical practice.

3.     The discussion could delve deeper into the limitations of the study, such as the exclusion of female participants and the use of fixed parameters in hypergraph construction.

4.     The authors address the discrepancy between the abstract's claim of "similar patterns of ALFF" and the results section's report of differences in ALFF patterns between datasets.

5.     The results section should specify the classification metric used to achieve the reported "over 80% accuracy" in classifying ASD subtypes.

6.     The claim in the discussion that this study is the "first attempt" to use high-order biological relationships for parsing heterogeneity in brain disorders should be substantiated or revised based on a thorough literature review.

7.       The manuscript could benefit from a final proofreading to correct minor grammatical errors and typos.

Comments on the Quality of English Language

Extensive editing of English language required

Round 2

Reviewer 2 Report

Comments and Suggestions for Authors

The manuscript was subjected to significant revisions in response to feedback received, improving both the clarity and the presentation of the experimental results.